# Spectral Representations for Convolutional Neural Networks

**Oren Rippel**
Department of Mathematics
Massachusetts Institute of Technology
rippel@math.mit.edu

**Jasper Snoek**
Twitter and Harvard SEAS
jsnoek@seas.harvard.edu

**Ryan P. Adams**
Twitter and Harvard SEAS
rpa@seas.harvard.edu

## Abstract

Discrete Fourier transforms provide a significant speedup in the computation of convolutions in deep learning. In this work, we demonstrate that, beyond its advantages for efficient computation, the spectral domain also provides a powerful representation in which to model and train convolutional neural networks (CNNs).

We employ spectral representations to introduce a number of innovations to CNN design. First, we propose *spectral pooling*, which performs dimensionality reduction by truncating the representation in the frequency domain. This approach preserves considerably more information per parameter than other pooling strategies and enables flexibility in the choice of pooling output dimensionality. This representation also enables a new form of stochastic regularization by randomized modification of resolution. We show that these methods achieve competitive results on classification and approximation tasks, without using any dropout or max-pooling.

Finally, we demonstrate the effectiveness of complex-coefficient spectral parameterization of convolutional filters. While this leaves the underlying model unchanged, it results in a representation that greatly facilitates optimization. We observe on a variety of popular CNN configurations that this leads to significantly faster convergence during training.

## 1  Introduction

Convolutional neural networks (CNNs) (LeCun et al., 1989) have been used to achieve unparalleled results across a variety of benchmark machine learning problems, and have been applied successfully throughout science and industry for tasks such as large scale image and video classification (Krizhevsky et al., 2012; Karpathy et al., 2014). One of the primary challenges of CNNs, however, is the computational expense necessary to train them. In particular, the efficient implementation of convolutional kernels has been a key ingredient of any successful use of CNNs at scale.

Due to its efficiency and the potential for amortization of cost, the discrete Fourier transform has long been considered by the deep learning community to be a natural approach to fast convolution (Bengio & LeCun, 2007). More recently, Mathieu et al. (2013); Vasilache et al. (2014) have demonstrated that convolution can be computed significantly faster using discrete Fourier transforms than directly in the spatial domain, even for tiny filters. This computational gain arises from the convenient property of operator duality between convolution in the spatial domain and element-wise multiplication in the frequency domain.

In this work, we argue that the frequency domain offers more than a computational trick for convolution: it also provides a powerful representation for modeling and training CNNs. Frequency decomposition allows studying an input across its various length-scales of variation, and as such provides a natural framework for the analysis of data with spatial coherence. We introduce two

applications of spectral representations. These contributions can be applied independently of each other.

**Spectral parametrization**   We propose the idea of learning the filters of CNNs directly in the frequency domain. Namely, we parametrize them as maps of complex numbers, whose discrete Fourier transforms correspond to the usual filter representations in the spatial domain.

Because this mapping corresponds to unitary transformations of the filters, this reparametrization does not alter the underlying model. However, we argue that the spectral representation provides an appropriate domain for parameter optimization, as the frequency basis captures typical filter structure well. More specifically, we show that filters tend to be considerably sparser in their spectral representations, thereby reducing the redundancy that appears in spatial domain representations. This provides the optimizer with more meaningful axis-aligned directions that can be taken advantage of with standard element-wise preconditioning.

We demonstrate the effectiveness of this reparametrization on a number of CNN optimization tasks, converging 2-5 times faster than the standard spatial representation.

**Spectral pooling**   Pooling refers to dimensionality reduction used in CNNs to impose a capacity bottleneck and facilitate computation. We introduce a new approach to pooling we refer to as *spectral pooling*. It performs dimensionality reduction by projecting onto the frequency basis set and then truncating the representation.

This approach alleviates a number of issues present in existing pooling strategies. For example, while max pooling is featured in almost every CNN and has had great empirical success, one major criticism has been its poor preservation of information (Hinton, 2014b,a). This weakness is exhibited in two ways. First, along with other stride-based pooling approaches, it implies a very sharp dimensionality reduction by at least a factor of 4 every time it is applied on two-dimensional inputs. Moreover, while it encourages translational invariance, it does not utilize its capacity well to reduce approximation loss: the maximum value in each window only reflects very local information, and often does not represent well the contents of the window.

In contrast, we show that spectral pooling preserves considerably more information for the same number of parameters. It achieves this exploiting the non-uniformity of typical inputs in their signal-to-noise ratio as a function of frequency. For example, natural images are known to have an expected power spectrum that follows an inverse power law: power is heavily concentrated in the lower frequencies — while higher frequencies tend to encode noise (Torralba & Oliva, 2003). As such, the elimination of higher frequencies in spectral pooling not only does minimal damage to the information in the input, but can even be viewed as a type of denoising.

In addition, spectral pooling allows us to specify any arbitrary output map dimensionality. This permits reduction of the map dimensionality in a slow and controlled manner as a function of network depth. Also, since truncation of the frequency representation exactly corresponds to reduction in resolution, we can supplement spectral pooling with stochastic regularization in the form of randomized resolution.

Spectral pooling can be implemented at a negligible additional computational cost in convolutional neural networks that employ FFT for convolution kernels, as it only requires matrix truncation. We also note that these two ideas are both compatible with the recently-introduced method of batch normalization (Ioffe & Szegedy, 2015), permitting even better training efficiency.

## 2   The Discrete Fourier Transform

The discrete Fourier transform (DFT) is a powerful way to decompose a spatiotemporal signal. In this section, we provide an introduction to a number of components of the DFT drawn upon in this work. We confine ourselves to the two-dimensional DFT, although all properties and results presented can be easily extended to other input dimensions.

Given an input $\mathbf{x} \in \mathbb{C}^{M \times N}$ (we address the constraint of real inputs in Subsection 2.1), its 2D DFT $\mathscr{F}(\mathbf{x}) \in \mathbb{C}^{M \times N}$ is given by

$$\mathscr{F}(\mathbf{x})_{hw} = \frac{1}{\sqrt{MN}} \sum_{m=0}^{M-1} \sum_{n=0}^{N-1} \mathbf{x}_{mn} e^{-2\pi i (\frac{mh}{M} + \frac{nw}{N})} \qquad \forall h \in \{0, \ldots, M-1\}, \forall w \in \{0, \ldots, N-1\} \, .$$

The DFT is linear and unitary, and so its inverse transform is given by $\mathscr{F}^{-1}(\cdot) = \mathscr{F}(\cdot)^*$, namely the conjugate of the transform itself.

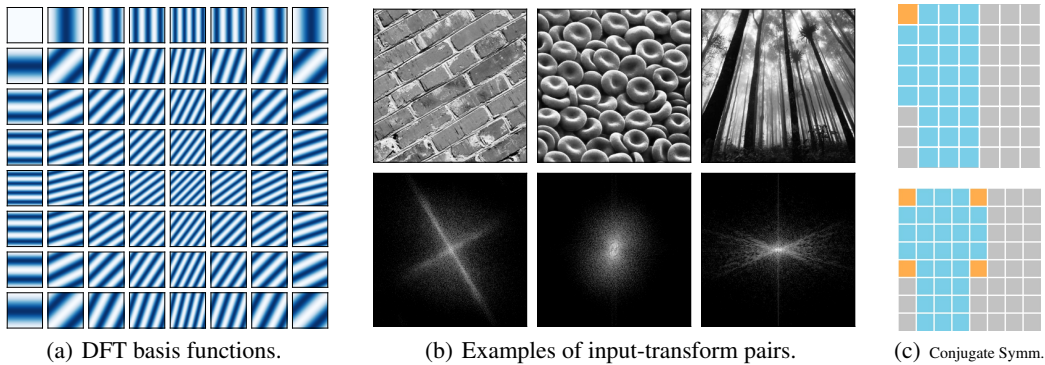

(a) DFT basis functions.     (b) Examples of input-transform pairs.     (c) Conjugate Symm.

Figure 1: Properties of discrete Fourier transforms. **(a)** All discrete Fourier basis functions of map size $8 \times 8$. Note the equivalence of some of these due to conjugate symmetry. **(b)** Examples of input images and their frequency representations, presented as log-amplitudes. The frequency maps have been shifted to center the DC component. Rays in the frequency domain correspond to spatial domain edges aligned perpendicular to these. **(c)** Conjugate symmetry patterns for inputs with odd (top) and even (bottom) dimensionalities. **Orange**: real-valuedness constraint. **Blue**: no constraint. **Gray**: value fixed by conjugate symmetry.

Intuitively, the DFT coefficients resulting from projections onto the different frequencies can be thought of as measures of correlation of the input with basis functions of various length-scales. See Figure 1(a) for a visualization of the DFT basis functions, and Figure 1(b) for examples of input-frequency map pairs.

The widespread deployment of the DFT can be partially attributed to the development of the Fast Fourier Transform (FFT), a mainstay of signal processing and a standard component of most math libraries. The FFT is an efficient implementation of the DFT with time complexity $\mathcal{O}\left(MN\log\left(MN\right)\right)$.

**Convolution using DFT** One powerful property of frequency analysis is the operator duality between convolution in the spatial domain and element-wise multiplication in the spectral domain. Namely, given two inputs $\mathbf{x}, \mathbf{f} \in \mathbb{R}^{M \times N}$, we may write

$$\mathscr{F}(\mathbf{x} * \mathbf{f}) = \mathscr{F}(\mathbf{x}) \odot \mathscr{F}(\mathbf{f}) \tag{1}$$

where by $*$ we denote a convolution and by $\odot$ an element-wise product.

**Approximation error** The unitarity of the Fourier basis makes it convenient for the analysis of approximation loss. More specifically, Parseval's Theorem links the $\ell_2$ loss between any input $\mathbf{x}$ and its approximation $\hat{\mathbf{x}}$ to the corresponding loss in the frequency domain:

$$\|\mathbf{x} - \hat{\mathbf{x}}\|_2^2 = \|\mathscr{F}(\mathbf{x}) - \mathscr{F}(\hat{\mathbf{x}})\|_2^2 . \tag{2}$$

An equivalent statement also holds for the inverse DFT operator. This allows us to quickly assess how an input is affected by any distortion we might make to its frequency representation.

## 2.1 Conjugate symmetry constraints

In the following sections of the paper, we will propagate signals and their gradients through DFT and inverse DFT layers. In these layers, we will represent the frequency domain in the complex field. However, for all layers apart from these, we would like to ensure that both the signal and its gradient are constrained to the reals. A necessary and sufficient condition to achieve this is *conjugate symmetry* in the frequency domain. Namely, for any transform $\mathbf{y} = \mathscr{F}(\mathbf{x})$ of some input $\mathbf{x}$, it must hold that

$$y_{mn} = y_{(M-m)\,\mathrm{mod}\,M,(N-n)\,\mathrm{mod}\,N}^* \qquad \forall m \in \{0, \ldots, M-1\}, \forall n \in \{0, \ldots, N-1\} . \tag{3}$$

Thus, intuitively, given the left half of our frequency map, the diminished number of degrees of freedom allows us to reconstruct the right. In effect, this allows us to store approximately half the parameters that would otherwise be necessary. Note, however, that this does not reduce the effective dimensionality, since each element consists of real and imaginary components. The conjugate symmetry constraints are visualized in Figure 1(c). Given a real input, its DFT will necessarily meet these. This symmetry can be observed in the frequency representations of the examples in Figure 1(b). However, since we seek to optimize over parameters embedded directly in the frequency domain, we need to pay close attention to ensure the conjugate symmetry constraints are enforced upon inversion back to the spatial domain (see Subsection 2.2).

## 2.2 Differentiation

Here we discuss how to propagate the gradient through a Fourier transform layer. This analysis can be similarly applied to the inverse DFT layer. Define $\mathbf{x} \in \mathbb{R}^{M \times N}$ and $\mathbf{y} = \mathscr{F}(\mathbf{x})$ to be the input and output of a DFT layer respectively, and $R : \mathbb{R}^{M \times N} \to \mathbb{R}$ a real-valued loss function applied to $\mathbf{y}$ which can be considered as the remainder of the forward pass. Since the DFT is a linear operator, its gradient is simply the transformation matrix itself. During back-propagation, then, this gradient is conjugated, and this, by DFT unitarity, corresponds to the application of the inverse transform:

$$\frac{\partial R}{\partial \mathbf{x}} = \mathscr{F}^{-1}\left(\frac{\partial R}{\partial \mathbf{y}}\right) \ . \tag{4}$$

There is an intricacy that makes matters a bit more complicated. Namely, the conjugate symmetry condition discussed in Subsection 2.1 introduces redundancy. Inspecting the conjugate symmetry constraints in Equation (3), we note their enforcement of the special case $y_{00} \in \mathbb{R}$ for $N$ odd, and $y_{00}, y_{\frac{N}{2},0}, y_{0,\frac{N}{2}}, y_{\frac{N}{2},\frac{N}{2}} \in \mathbb{R}$ for $N$ even. For all other indices they enforce conjugate equality of pairs of distinct elements. These conditions imply that the number of unconstrained parameters is about half the map in its entirety.

# 3 Spectral Pooling

The choice of a pooling technique boils down to the selection of an appropriate set of basis functions to project onto, and some truncation of this representation to establish a lower-dimensionality approximation to the original input. The idea behind spectral pooling stems from the observation that the frequency domain provides an ideal basis for inputs with spatial structure. We first discuss the technical details of this approach, and then its advantages.

Spectral pooling is straightforward to understand and to implement. We assume we are given an input $\mathbf{x} \in \mathbb{R}^{M \times N}$, and some desired output map dimensionality $H \times W$. First, we compute the discrete Fourier transform of the input into the frequency domain as $\mathbf{y} = \mathscr{F}(\mathbf{x}) \in \mathbb{C}^{M \times N}$, and assume that the DC component has been shifted to the center of the domain as is standard practice. We then crop the frequency representation by maintaining only the central $H \times W$ submatrix of frequencies, which we denote as $\hat{\mathbf{y}} \in \mathbb{C}^{H \times W}$. Finally, we map this approximation back into the spatial domain by taking its inverse DFT as $\hat{\mathbf{x}} = \mathscr{F}^{-1}(\hat{\mathbf{y}}) \in \mathbb{R}^{H \times W}$. These steps are listed in Algorithm 1. Note that some of the conjugate symmetry special cases described in Subsection 2.2 might be broken by this truncation. As such, to ensure that $\hat{\mathbf{x}}$ is real-valued, we must treat these individually with TREATCORNERCASES, which can be found in the supplementary material.

Figure 2 demonstrates the effect of this pooling for various choices of $H \times W$. The back-propagation procedure is quite intuitive, and can be found in Algorithm 2 (REMOVEREDUNDANCY and RECOVERMAP can be found in the supplementary material). In Subsection 2.2, we addressed the nuances of differentiating through DFT and inverse DFT layers. Apart from these, the last component left undiscussed is differentiation through the truncation of the frequency matrix, but this corresponds to a simple zero-padding of the gradient maps to the appropriate dimensions.

In practice, the DFTs are the computational bottlenecks of spectral pooling. However, we note that in convolutional neural networks that employ FFTs for convolution computation, spectral pooling can be implemented at a negligible additional computational cost, since the DFT is performed regardless.

We proceed to discuss a number of properties of spectral pooling, which we then test comprehensively in Section 5.

---

| Algorithm 1: Spectral pooling |
| --- |
| **Input:** Map $\mathbf{x} \in \mathbb{R}^{M \times N}$, output size $H \times W$ |
| **Output:** Pooled map $\hat{\mathbf{x}} \in \mathbb{R}^{H \times W}$ |
| 1: $\mathbf{y} \leftarrow \mathscr{F}(\mathbf{x})$ |
| 2: $\hat{\mathbf{y}} \leftarrow$ CROPSPECTRUM$(\mathbf{y}, H \times W)$ |
| 3: $\hat{\mathbf{y}} \leftarrow$ TREATCORNERCASES$(\hat{\mathbf{y}})$ |
| 4: $\hat{\mathbf{x}} \leftarrow \mathscr{F}^{-1}(\hat{\mathbf{y}})$ |

| Algorithm 2: Spectral pooling back-propagation |
| --- |
| **Input:** Gradient w.r.t output $\frac{\partial R}{\partial \hat{\mathbf{x}}}$ |
| **Output:** Gradient w.r.t input $\frac{\partial R}{\partial \mathbf{x}}$ |
| 1: $\hat{\mathbf{z}} \leftarrow \mathscr{F}\left(\frac{\partial R}{\partial \hat{\mathbf{x}}}\right)$ |
| 2: $\hat{\mathbf{z}} \leftarrow$ REMOVEREDUNDANCY$(\hat{\mathbf{z}})$ |
| 3: $\mathbf{z} \leftarrow$ PADSPECTRUM$(\hat{\mathbf{z}}, M \times N)$ |
| 4: $\mathbf{z} \leftarrow$ RECOVERMAP$(\mathbf{z})$ |
| 5: $\frac{\partial R}{\partial \mathbf{x}} \leftarrow \mathscr{F}^{-1}(\mathbf{z})$ |

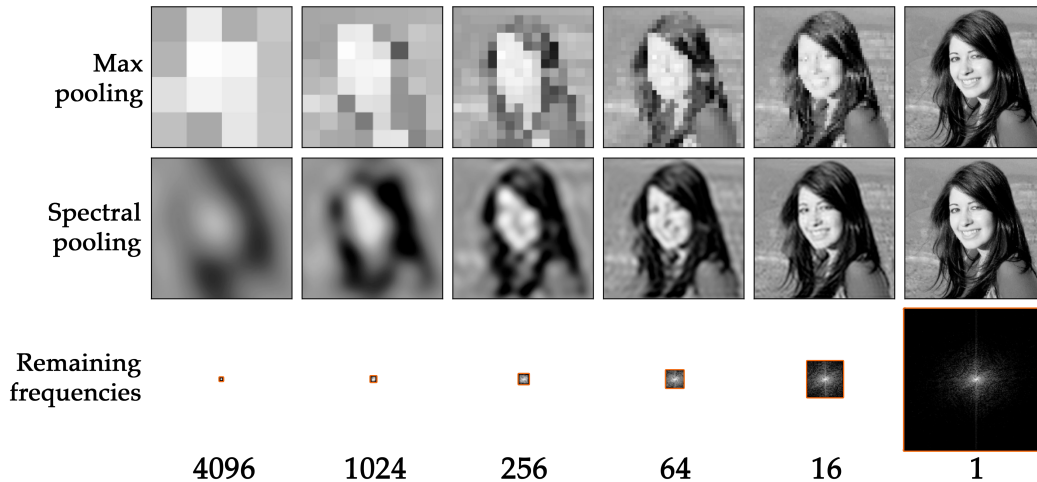

Figure 2: Approximations for different pooling schemes, for different factors of dimensionality reduction. Spectral pooling projects onto the Fourier basis and truncates it as desired. This retains significantly more information and permits the selection of any arbitrary output map dimensionality.

### 3.1 Information preservation

Spectral pooling can significantly increase the amount of retained information relative to max-pooling in two distinct ways. First, its representation maintains more information for the same number of degrees of freedom. Spectral pooling reduces the information capacity by tuning the resolution of the input precisely to match the desired output dimensionality. This operation can also be viewed as linear low-pass filtering and it exploits the non-uniformity of the spectral density of the data with respect to frequency. That is, that the power spectra of inputs with spatial structure, such as natural images, carry most of their mass on lower frequencies. As such, since the amplitudes of the higher frequencies tend to be small, Parseval's theorem from Section 2 informs us that their elimination will result in a representation that minimizes the $\ell_2$ distortion after reconstruction.

Second, spectral pooling does not suffer from the sharp reduction in output dimensionality exhibited by other pooling techniques. More specifically, for stride-based pooling strategies such as max pooling, the number of degrees of freedom of two-dimensional inputs is reduced by at least 75% as a function of stride. In contrast, spectral pooling allows us to specify any arbitrary output dimensionality, and thus allows us to reduce the map size gradually as a function of layer.

### 3.2 Regularization via resolution corruption

We note that the low-pass filtering radii, say $R_H$ and $R_W$, can be chosen to be smaller than the output map dimensionalities $H, W$. Namely, while we truncate our input frequency map to size $H \times W$, we can further zero-out all frequencies outside the central $R_H \times R_W$ square. While this maintains the output dimensionality $H \times W$ of the input domain after applying the inverse DFT, it effectively reduces the resolution of the output. This can be seen in Figure 2.

This allows us to introduce regularization in the form of random resolution reduction. We apply this stochastically by assigning a distribution $p_R(\cdot)$ on the frequency truncation radius (for simplicity we apply the same truncation on both axes), sampling from this a random radius at each iteration, and wiping out all frequencies outside the square of that size. Note that this can be regarded as an application of nested dropout (Rippel et al., 2014) on both dimensions of the frequency decomposition of our input. In practice, we have had success choosing $p_R(\cdot) = U_{[H_{\min}, H]}(\cdot)$, i.e., a uniform distribution stretching from some minimum value all the way up to the highest possible resolution.

## 4 Spectral Parametrization of CNNs

Here we demonstrate how to learn the filters of CNNs directly in their frequency domain representations. This offers significant advantages over the traditional spatial representation, which we show empirically in Section 5.

Let us assume that for some layer of our convolutional neural network we seek to learn filters of size $H \times W$. To do this, we parametrize each filter $\mathbf{f} \in \mathbb{C}^{H \times W}$ in our network directly in the frequency domain. To attain its spatial representation, we simply compute its inverse DFT

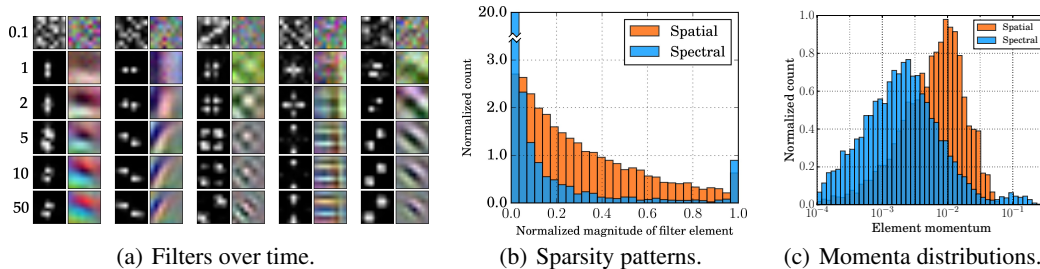

(a) Filters over time.  (b) Sparsity patterns.  (c) Momenta distributions.

Figure 3: Learning dynamics of CNNs with spectral parametrization. The histograms have been produced after 10 epochs of training on CIFAR-10 by each method, but are similar throughout. **(a)** Progression over several epochs of filters parametrized in the frequency domain. Each pair of columns corresponds to the spectral parametrization of a filter and its inverse transform to the spatial domain. Filter representations tend to be more local in the Fourier basis. **(b)** Sparsity patterns for the different parametrizations. Spectral representations tend to be considerably sparser. **(c)** Distributions of momenta across parameters for CNNs trained with and without spectral parametrization. In the spectral parametrization considerably fewer parameters are updated.

as $\mathscr{F}^{-1}(\mathbf{f}) \in \mathbb{R}^{H \times W}$. From this point on, we proceed as we would for any standard CNN by computing the convolution of the filter with inputs in our mini-batch, and so on.

The back-propagation through the inverse DFT is virtually identical to the one of spectral pooling described in Section 3. We compute the gradient as outlined in Subsection 2.2, being careful to obey the conjugate symmetry constraints discussed in Subsection 2.1.

We emphasize that this approach does not change the underlying CNN model in any way — only the way in which it is parametrized. Hence, this only affects the way the solution space is explored by the optimization procedure.

## 4.1 Leveraging filter structure

This idea exploits the observation that CNN filters have a very characteristic structure that reappears across data sets and problem domains. That is, CNN weights can typically be captured with a small number of degrees of freedom. Represented in the spatial domain, however, this results in significant redundancy.

The frequency domain, on the other hand, provides an appealing basis for filter representation: characteristic filters (e.g., Gabor filters) are often very localized in their spectral representations. This follows from the observation that filters tend to feature very specific length-scales and orientations. Hence, they tend to have nonzero support in a narrow set of frequency components. This hypothesis can be observed qualitatively in Figure 3(a) and quantitatively in Figure 3(b).

Empirically, in Section 5 we observe that spectral representations of filters leads to a convergence speedup by 2-5 times. We remark that, had we trained our network with standard stochastic gradient descent, the linearity of differentiation and parameter update would have resulted in exactly the same filters regardless of whether they were represented in the spatial or frequency domain during training (this is true for *any* invertible linear transformation of the parameter space).

However, as discussed, this parametrization corresponds to a rotation to a more meaningful axis alignment, where the number of relevant elements has been significantly reduced. Since modern optimizers implement update rules that consist of adaptive element-wise rescaling, they are able to leverage this axis alignment by making large updates to a small number of elements. This can be seen quantitatively in Figure 3(c), where the optimizer — Adam (Kingma & Ba, 2015), in this case — only touches a small number of elements in its updates.

There exist a number of extensions of the above approach we believe would be quite promising in future work; we elaborate on these in the discussion.

## 5 Experiments

We demonstrate the effectiveness of spectral representations in a number of different experiments. We ran all experiments on code optimized for the Xeon Phi coprocessor. We used Spearmint (Snoek et al., 2015) for Bayesian optimization of hyperparameters with 5-20 concurrent evaluations.

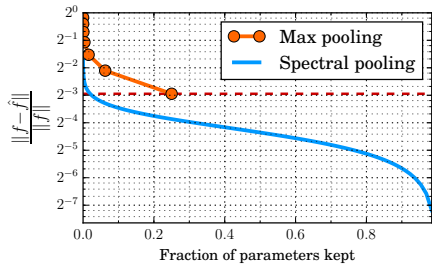

| Method | CIFAR-10 | CIFAR-100 |
|---|---|---|
| Stochastic pooling | 15.13% | 41.51% |
| Maxout | 11.68% | 38.57% |
| Network-in-network | 10.41% | 35.68% |
| Deeply supervised | 9.78% | 34.57% |
| **Spectral pooling** | **8.6%** | **31.6%** |

(a) Approximation loss for the ImageNet validation set.

(b) Classification rates.

Figure 4: **(a)** Average information dissipation for the ImageNet validation set as a function of fraction of parameters kept. This is measured in $\ell_2$ error normalized by the input norm. The red horizontal line indicates the best error rate achievable by max pooling. **(b)** Test errors on CIFAR-10/100 without data augmentation of the optimal spectral pooling architecture, as compared to current state-of-the-art approaches: stochastic pooling (Zeiler & Fergus, 2013), Maxout (Goodfellow et al., 2013), network-in-network (Lin et al., 2013), and deeply-supervised nets (Lee et al., 2014).

## 5.1 Spectral pooling

**Information preservation** We test the information retainment properties of spectral pooling on the validation set of ImageNet (Russakovsky et al., 2015). For the different pooling strategies we plot the average approximation loss resulting from pooling to different dimensionalities. This can be seen in Figure 4. We observe the two aspects discussed in Subsection 3.1: first, spectral pooling permits significantly better reconstruction for the same number of parameters. Second, for max pooling, the only knob controlling the coarseness of approximation is the stride, which results in severe quantization and a constraining lower bound on preserved information (marked in the figure as a horizontal red line). In contrast, spectral pooling permits the selection of any output dimensionality, thereby producing a smooth curve over all frequency truncation choices.

**Classification with convolutional neural networks** We test spectral pooling on different classification tasks. We hyperparametrize and optimize the following CNN architecture:

$$\left( \mathrm{C}_{3\times3}^{96+32m} \rightarrow \mathrm{SP}_{\downarrow\lfloor\gamma H_m\rfloor\times\lfloor\gamma H_m\rfloor} \right)_{m=1}^{M} \rightarrow \mathrm{C}_{1\times1}^{96+32M} \rightarrow \mathrm{C}_{1\times1}^{10/100} \rightarrow \mathrm{GA} \rightarrow \mathrm{Softmax} \tag{5}$$

Here, by $\mathrm{C}_S^F$ we denote a convolutional layer with $F$ filters each of size $S$, by $\mathrm{SP}_{\downarrow S}$ a spectral pooling layer with output dimensionality $S$, and GA the global averaging layer described in Lin et al. (2013). We upper-bound the number of filters per layer as 288. Every convolution and pooling layer is followed by a ReLU nonlinearity. We let $H_m$ be the height of the map of layer $m$. Hence, each spectral pooling layer reduces each output map dimension by factor $\gamma \in (0,1)$. We assign frequency dropout distribution $p_R(\cdot; m, \alpha, \beta) = U_{[\lfloor c_m H_m\rfloor, H_m]}(\cdot)$ for layer $m$, total layers $M$ and with $c_m(\alpha, \beta) = \alpha + \frac{m}{M}(\beta - \alpha)$ for some constants $\alpha, \beta \in \mathbb{R}$. This parametrization can be thought of as some linear parametrization of the dropout rate as a function of the layer.

We perform hyperparameter optimization on the dimensionality decay rate $\gamma \in [0.25, 0.85]$, number of layers $M \in \{1, \ldots, 15\}$, resolution randomization hyperparameters $\alpha, \beta \in [0, 0.8]$, weight decay rate in $[10^{-5}, 10^{-2}]$, momentum in $[1 - 0.1^{0.5}, 1 - 0.1^2]$ and initial learning rate in $[0.1^4, 0.1]$. We train each model for 150 epochs and anneal the learning rate by a factor of 10 at epochs 100 and 140. We intentionally use no dropout nor data augmentation, as these introduce a number of additional hyperparameters which we want to disambiguate as alternative factors for success.

Perhaps unsurprisingly, the optimal hyperparameter configuration assigns the slowest possible layer map decay rate $\gamma = 0.85$. It selects randomized resolution reduction constants of about $\alpha \approx 0.30, \beta \approx 0.15$, momentum of about 0.95 and initial learning rate 0.0088. These settings allow us to attain classification rates of 8.6% on CIFAR-10 and 31.6% on CIFAR-100. These are competitive results among approaches that do not employ data augmentation: a comparison to state-of-the-art approaches from the literature can be found in Table 4(b).

## 5.2 Spectral parametrization of CNNs

We demonstrate the effectiveness of spectral parametrization on a number of CNN optimization tasks, for different architectures and for different filter sizes. We use the notation $\mathrm{MP}_S^T$ to denote a max pooling layer with size $S$ and stride $T$, and $\mathrm{FC}^F$ is a fully-connected layer with $F$ filters.

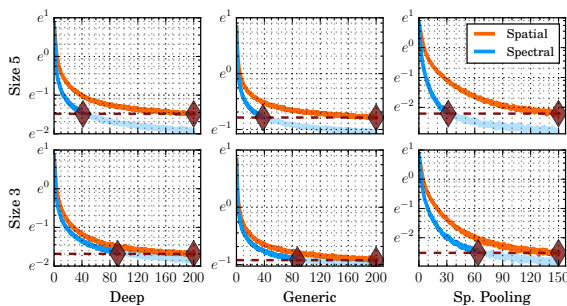

| Architecture | Filter size | Speedup factor |
|---|---|---|
| Deep (7) | $3 \times 3$ | 2.2 |
| Deep (7) | $5 \times 5$ | 4.8 |
| Generic (6) | $3 \times 3$ | 2.2 |
| Generic (6) | $5 \times 5$ | 5.1 |
| Sp. Pooling (5) | $3 \times 3$ | 2.4 |
| Sp. Pooling (5) | $5 \times 5$ | 4.8 |

(a) Training curves.

(b) Speedup factors.

Figure 5: Optimization of CNNs via spectral parametrization. All experiments include data augmentation. **(a)** Training curves for the various experiments. The remainder of the optimization past the matching point is marked in light blue. The red diamonds indicate the relative epochs in which the asymptotic error rate of the spatial approach is achieved. **(b)** Speedup factors for different architectures and filter sizes. A non-negligible speedup is observed even for tiny $3 \times 3$ filters.

The first architecture is the generic one used in a variety of deep learning papers, such as Krizhevsky et al. (2012); Snoek et al. (2012); Krizhevsky (2009); Kingma & Ba (2015):

$$C_{3\times3}^{96} \rightarrow MP_{3\times3}^{2} \rightarrow C_{3\times3}^{192} \rightarrow MP_{3\times3}^{2} \rightarrow FC^{1024} \rightarrow FC^{512} \rightarrow \text{Softmax} \tag{6}$$

The second architecture we consider is the one employed in Snoek et al. (2015), which was shown to attain competitive classification rates. It is deeper and more complex:

$$c_{3\times3}^{96} \rightarrow c_{3\times3}^{96} \rightarrow MP_{3\times3}^{2} \rightarrow c_{3\times3}^{192} \rightarrow c_{3\times3}^{192} \rightarrow c_{3\times3}^{192} \rightarrow MP_{3\times3}^{2} \rightarrow c_{1\times1}^{192} \rightarrow c_{1\times1}^{10/100} \rightarrow GA \rightarrow \text{Softmax} \tag{7}$$

The third architecture considered is the spectral pooling network from Equation 5. To increase the difficulty of optimization and reflect real training conditions, we supplemented all networks with data augmentation in the form of translations, horizontal reflections, HSV perturbations and dropout.

We initialized both spatial and spectral filters in the spatial domain as the same values; for the spectral parametrization experiments we then computed the Fourier transform of these to attain their frequency representations. We optimized all networks using the Adam (Kingma & Ba, 2015) update rule, a variant of RMSprop that we find to be a fast and robust optimizer.

The training curves can be found in Figure 5(a) and the respective factors of convergence speedup in Table 5. Surprisingly, we observe non-negligible speedup even for tiny filters of size $3 \times 3$, where we did not expect the frequency representation to have much room to exploit spatial structure.

# 6   Discussion and remaining open problems

In this work, we demonstrated that spectral representations provide a rich spectrum of applications. We introduced spectral pooling, which allows pooling to any desired output dimensionality while retaining significantly more information than other pooling approaches. In addition, we showed that the Fourier functions provide a suitable basis for filter parametrization, as demonstrated by faster convergence of the optimization procedure.

One possible future line of work is to embed the network in its entirety in the frequency domain. In models that employ Fourier transforms to compute convolutions, at every convolutional layer the input is FFT-ed and the elementwise multiplication output is then inverse FFT-ed. These back-and-forth transformations are very computationally intensive, and as such it would be desirable to strictly remain in the frequency domain. However, the reason for these repeated transformations is the application of nonlinearities in the forward domain: if one were to propose a sensible nonlinearity in the frequency domain, this would spare us from the incessant domain switching.

**Acknowledgements**   We would like to thank Prabhat, Michael Gelbart and Matthew Johnson for useful discussions and assistance throughout this project. Jasper Snoek was a fellow in the Harvard Center for Research on Computation and Society. This work is supported by the Applied Mathematics Program within the Office of Science Advanced Scientific Computing Research of the U.S. Department of Energy under contract No. DE-AC02-05CH11231. This work used resources of the National Energy Research Scientific Computing Center (NERSC). We thank Helen He and Doug Jacobsen for providing us with access to the Babbage Xeon-Phi testbed at NERSC.

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
