[Supplementary Material · supplementary_nips.pdf]

# Spectral Representations for Convolutional Neural Networks: Supplementary Material

## 1 Appendix A: Algorithmic Implementation Details

Here we provide additional detail pertaining to the specific algorithmic implementation of the spectral pooling and spectral parameterization. Algorithms 1 and 2 detail the steps required to compute the spectral pooling and corresponding back-propagation respectively. CROPSPECTRUM and PADSPECTRUM are self-explanatory: they crop or zero-pad the frequency spectrum to the appropriate dimensionalities, respectively.

---

**Algorithm 1: Spectral pooling**

**Input:** Map $\mathbf{x} \in \mathbb{R}^{M \times N}$, output size $H \times W$
**Output:** Pooled map $\hat{\mathbf{x}} \in \mathbb{R}^{H \times W}$
1: $\mathbf{y} \leftarrow \mathscr{F}(\mathbf{x})$
2: $\hat{\mathbf{y}} \leftarrow \text{CROPSPECTRUM}(\mathbf{y}, H \times W)$
3: $\hat{\mathbf{y}} \leftarrow \text{TREATCORNERCASES}(\hat{\mathbf{y}})$
4: $\hat{\mathbf{x}} \leftarrow \mathscr{F}^{-1}(\hat{\mathbf{y}})$

---

**Algorithm 2: Spectral pooling back-propagation**

**Input:** Gradient w.r.t output $\frac{\partial R}{\partial \hat{\mathbf{x}}}$
**Output:** Gradient w.r.t input $\frac{\partial R}{\partial \mathbf{x}}$
1: $\hat{\mathbf{z}} \leftarrow \mathscr{F}\left(\frac{\partial R}{\partial \hat{\mathbf{x}}}\right)$
2: $\hat{\mathbf{z}} \leftarrow \text{REMOVEREDUNDANCY}(\hat{\mathbf{z}})$
3: $\mathbf{z} \leftarrow \text{PADSPECTRUM}(\hat{\mathbf{z}}, M \times N)$
4: $\mathbf{z} \leftarrow \text{RECOVERMAP}(\mathbf{z})$
5: $\frac{\partial R}{\partial \mathbf{x}} \leftarrow \mathscr{F}^{-1}(\mathbf{z})$

---

**Algorithm 3: TREATCORNERCASES**

**Input:** Input map $\mathbf{y} \in \mathbb{C}^{M \times N}$
**Output:** Output map $\mathbf{z}$ with corner cases obeying conjugate symmetry, special case indices $S$
1: $\mathbf{z} \leftarrow \mathbf{y}$
2: $S \leftarrow \{(0,0)\}$
3: **if** $M$ is even **then**
4:     $S \leftarrow \{(\frac{M}{2}, 0)\}$
5: **end if**
6: **if** $N$ is even **then**
7:     $S \leftarrow \{(0, \frac{N}{2})\}$
8: **end if**
9: **if** $M$ is even and $N$ is even **then**
10:     $S \leftarrow \{(\frac{M}{2}, \frac{N}{2})\}$
11: **end if**
12: **for** $i \in S$ **do**
13:     $\text{Im}(\mathbf{z}_i) \leftarrow 0$
14: **end for**

---

---
Algorithm 4: REMOVEREDUNDANCY
---

**Input:** Input gradient map $\mathbf{y} \in \mathbb{C}^{M \times N}$
**Output:** Gradient $\mathbf{z}$ in terms of unconstrained parameters only
1: $\mathbf{z}, S \leftarrow \textsc{TreatCornerCases}(\mathbf{y})$
2: $I \leftarrow \emptyset$
3: **for** $m = 0, \ldots, M-1$ **do**
4:    **for** $n = 0, \ldots, \lfloor \frac{N}{2} \rfloor$ **do**
5:       **if** $(m,n) \notin S$ **then**
6:          **if** $(m,n) \notin I$ **then**
7:             $\mathbf{z}_{m,n} \leftarrow 2\mathbf{z}_{m,n}$
8:             $I \leftarrow I \cup \{(m,n), ((M-m)\operatorname{mod}M, (N-n)\operatorname{mod}N)\}$
9:          **else**
10:            $\mathbf{z}_{m,n} \leftarrow 0$
11:          **end if**
12:       **end if**
13:    **end for**
14: **end for**

---
Algorithm 5: RECOVERMAP
---

**Input:** Input gradient $\mathbf{y} \in \mathbb{C}^{M \times N}$ parametrized by unconstrained elements only
**Output:** Full gradient $\mathbf{z}$ with recovered redundancy
1: $\mathbf{z}, S \leftarrow \textsc{TreatCornerCases}(\mathbf{y})$
2: $I \leftarrow \emptyset$
3: **for** $m = 0, \ldots, M-1$ **do**
4:    **for** $n = 0, \ldots, \lfloor \frac{N}{2} \rfloor$ **do**
5:       **if** $(m,n) \notin S$ **then**
6:          **if** $(m,n) \notin I$ **then**
7:             $\mathbf{z}_{m,n} \leftarrow \frac{1}{2}\mathbf{z}_{m,n}$
8:             $\mathbf{z}_{(M-m)\operatorname{mod}M, (N-n)\operatorname{mod}N} \leftarrow \mathbf{z}_{m,n}$
9:             $I \leftarrow I \cup \{(m,n), ((M-m)\operatorname{mod}M, (N-n)\operatorname{mod}N)\}$
10:          **else**
11:            $\mathbf{z}_{m,n} \leftarrow 0$
12:          **end if**
13:       **end if**
14:    **end for**
15: **end for**