[Reviews · NeurIPS 2015]

Submitted by Assigned_Reviewer_1

The authors propose a spectral representation of CNN parameters (in terms of the discrete Fourier transformation) and describe a number of applications and benefits that leverage this parametrization. Specifically, they introduce spectral pooling, which essentially clips high frequency information allowing for better information preservation and arbitrary resolution, regularization (akin to nested dropout) by random resolution reduction, along with all of the algorithmic details necessary for implementation (e.g. backprop.)

Experimental results show many benefits, including increased filter sparsity, better information preservation during pooling, and faster training. These lead to competitive results on CIFAR-10/100 (among approaches that do not employ data augmentation) and significant training speedups, even with small filter sizes.

It would be interesting to see timing comparisons with methods that only perform convolutions in the spectral domain (while still representing parameters and back-propagating gradients in the spatial domain.) The authors claim that a spectral parametrization produces "more meaningful axis alignment" which could result in more efficient training with adaptive element-wise rescaling. However, it is not clear the magnitude of these gains in comparison to just frequency-domain convolution (which is known to be faster.)

Furthermore, some missing experiments include: tuning plots showing the sensitivity of the chosen hyper-parameters, accuracy comparisons between dropout and spectral pooling (instead of just approximation loss comparisons), and demonstrations on more challenging tasks (e.g. imagenet.)
Summary: This paper proposes a new parametrization of CNN model parameters based on complex frequency content. This is a natural representation that is commonly used for efficient convolution, but the authors demonstrate that it can be beneficial in other respects as well. While some more impactful details are missing (e.g. a spectral nonlinearity to avoid repeated DFT and inverse DFT operations) the approach is interesting and demonstrates potential for further work.

Submitted by Assigned_Reviewer_2

Paper proposes to do pooling in spectral rather than spatial domain.

I did not go through the paper closely enough to fully understand how this is done.

Results look impressive.
Summary: Seems a very interesting idea, and amazing if true, but I don't follow vision stuff closely enough to know how seriously to take it.

Submitted by Assigned_Reviewer_3

The authors propose to learn

Quality and Clarity:

No objections here, well written paper.

Originality:

There is a number of works that authors skipped in related work (which is entirely missing) and where spectral parametrisation was used and learned via backprop. This must be added and discussed so reader gets a clear message what's the real contribution of this work.

Unsupervised learning of visual invariance with temporal coherence, W, Zhou, Andrew Ng and Kai Yu, NIPS workshop 2013

Multilayer Neural Network based on Multi-Valued Neurons and the Blur Identification Problem, Aizenberg, I. ; Texas A&M Univ.-Texarkana, Texarkana ; Paliy, D. ; Astola, J.T.

Akira Hirose, Complex-valued neural networks

Significance:

Frequency domain is much more meaningful in solving multiple tasks, and

as such practical working example of one of possible ways to incorporate this in the model is important.
Summary: An interesting extension to complex-value parametrisations of neural networks (not novel) and spectral pooling (novel). Good paper as it makes it actually working.

Submitted by Assigned_Reviewer_4

Comments/questions about spectral pooling: - L212: "First [spectral pooling] representation maintains more information for the same number of degrees of freedom."

-- I disagree with this statement.

The amount of information in an image or its DFT is the same, after all it is just a linear transformation of the data.

More information is present because the output of spectral pooling is larger than typical max/avg-pooling.

In fact, avg-pooling is very similar to spectral pooling: it's a moving average filter (a different low-pass filter) followed by downsampling.

The same amount of info could be kept using max/avg-pooling by using overlapping windows (also related to the claim on L250 that max-pooling has a large reduction of dimensionality). - maxpooling is translation invariant, while spectral pooling is not translation invariant (translation is encoded in the phase of the DFT).

Supposedly translation invariance helps in recognition tasks.

Does spectral pooling get similar results to max-pooling on object recognition tasks (e.g., ImageNet tested in Fig 4a)? - given the similarity to avg-pooling, it would be nice to see a comparison on CIFAR.

Comments/questions about spectral parameterization: - for spectral parameterization, it would be interesting to see whether the re-parameterization allows better learning for smaller training set sizes. - L301 - Does the reparameterization affect the regularization of the weights?

I suppose not, since it is based on a unitary linear transform.

It is worth mentioning this.

- L319 - unitary transform is also a condition for the training to be the same. Otherwise regularization of the weights is affected. - L424-425 - The experiments show that fewer iterations are needed for the spectral representation.

But this ignores the time-cost for calculating the DFT and inverse DFT.

What is the speed-up in terms of actual clock time?

Summary: The paper proposes to use spectral DFT representation for parameters in a CNN: 1) DFT parameterization of the weights for easier learning; 2) spectral pooling, which is doing low-pass filtering in the DFT domain, downssampling, and then projecting back as a smaller image.

Experimental results show faster convergence (in terms of iterations) using spectral parameterization, and better classification using spectral pooling.

The idea is interesting, but there are a number of things that need to be addressed to improve the presentation and increase the impact.

Author Feedback
Author rebuttal: We would first like to thank our reviewers for their thoughtful and encouraging feedback. This will allow us to improve our paper and its presentation.

Reviewer 6, we are grateful for your valuable input. In your review, you noted that spectral pooling maintains the same amount of information as max pooling for the same output map dimensionality. We would like to clarify that spectral pooling utilizes the given degrees of freedom significantly better than max pooling in terms of its approximation of the input (which is our metric for assessing preservation of information), as can be seen in Figure 4(a). We of course agree that the DFT of an input contains the same amount of information as the input itself, but the statement we were specifically attempting to make is regarding the form of dimensionality reduction employed. That is, spectral pooling is much more suitable to be used on inputs with spatial coherence, as it its retainment of the lower frequencies allows it to exploit the fact that typical power spectra tend to be concentrated around these. In addition, you mentioned that the same reduction in dimensionality could be achieved using max/avg-pooling by using overlapping windows. This is not true: in the best case scenario for stride-based pooling methods, namely overlapping windows with stride 2 (stride 1 does not perform any dimensionality reduction), the output dimensionality can be at most 25% of that of the input. Spectral pooling, on the other hand, allow choosing any arbitrary output map size. Finally, we did test average pooling against spectral pooling and max pooling on CIFAR-10, but it was not nearly as competitive as these (which is also why it hasn't been popular in the community).

And to address your questions:
**Q: Does the re-parameterization affect the regularization of the weights?
**A: Exactly as you said, it does not affect regularization due to the unitarity of the transformation. This is certainly a point we should add to the paper.

**Q:What is the speed-up in terms of actual clock time?
**A: This touches on our reply to reviewer 1, so we would please like to refer you to the below. To add to this, this means that the relative gains in terms of wallclock are the same as the ones observed for convergence as function of number of iterations.

Reviewer 1, you mentioned that it is not clear whether the gains in training speed are due to faster convergence or due to faster convolution in the frequency domain. We would like to clarify that our results do not at all take into account training time, but are instead reported in terms of the number of iterations - and thus the gains are purely in terms of iterations to convergence. Note that since spectral re-parametrization only requires taking the FFT of (tiny!) filters, it adds negligible computational cost to the training procedure.

All the above concerns are certainly points we will clarify in the camera-ready version of the paper.